# Irradiation Alters the Expression of MUC1, CD44 and Hyaluronan in Oral Mucosal Epithelium

**DOI:** 10.3390/biomedicines10112816

**Published:** 2022-11-04

**Authors:** Bina Kashyap, Konsta Naumanen, Jopi Mikkonen, Hannah Dekker, Engelbert Schulten, Elisabeth Bloemena, Sanna Pasonen-Seppänen, Arja Kullaa

**Affiliations:** 1Institute of Dentistry, University of Eastern Finland, Kuopio Campus and Educational Dental Clinic, Kuopio University Hospital, P.O. Box 1627, 70211 Kuopio, Finland; 2Amsterdam UMC and Academic Centre for Dentistry Amsterdam (ACTA), Vrije Universiteit Amsterdam, Department of Oral and Maxillofacial Surgery/Oral Pathology, De Boelelaan 1117, 1081 Amsterdam, The Netherlands; 3Institute of Biomedicine, University of Eastern Finland, Kuopio Campus, P.O. Box 1627, 70211 Kuopio, Finland

**Keywords:** mucin 1, MUC1, CD44, hyaluronan, biomarkers, oral mucosa, oral cancer, radiotherapy

## Abstract

**Purpose**: It is well established that cancer cells exploit aberrant synthesis of mucin 1 (MUC1) and hyaluronan (HA) synthesis along with HA’s physiological cell surface receptor CD44. However, their role in irradiated oral tissue has not been reported previously. We, therefore, aimed to study MUC1, CD44 and HA immunohistochemically in irradiated oral mucosa and their role in the long-term effects after radiotherapy. **Materials and Methods**: Oral mucosal biopsies were obtained from healthy subjects as controls and from patients after radiotherapy for head and neck cancer (irradiated group) during dental implant surgery. The presence of MUC1, CD44, and HA in oral mucosa was studied by immunohistochemical methods. The differences in the localization and intensity in the oral epithelium between control and irradiated tissue were analyzed. **Results:** The staining intensity of MUC1 was confined to the superficial epithelial layer, whereas HA and CD44 were found in the cell membranes in the epithelial basal and intermediate layers of control specimens. In irradiated epithelium, MUC1 staining was distributed throughout all the layers of the oral epithelium, with significant staining in the basal and intermediate layers. Accordingly, HA and CD44 staining extended to involve the superficial cells of the irradiated epithelium. The staining pattern of MUC1 and CD44 showed significant changes in irradiated samples. **Conclusions:** Our results showed that the staining intensities of MUC1, CD44, and HA were significantly elevated in irradiated tissue compared to controls. MUC1, CD44, and HA are important markers and take part in long-term changes in the oral mucosa after radiotherapy.

## 1. Introduction

Head and neck cancers (HNC) constitute one of the most common cancers worldwide [1]. Of these, the most common histological subtype of head and neck squamous cell carcinoma arises from the mucosal surfaces of the oral cavity, oropharynx, hypopharynx, and larynx [2]. Oral squamous cell carcinomas (OSCC) can modify cellular functions and propagate altered cells towards migration, invasion, and metastasis, thus causing morbidity in patients. Surgery and radiotherapy (RT), chemotherapy, or their combination are the standard treatment options for both oral and other HNC cancers. Post-operative RT has been advocated for patients with a high risk of disease recurrence [3]. It is generally assumed that RT enhances the anti-tumor effect by modifying cellular and extracellular elements. However, it may exert potential tumor-promoting effects resulting in tumor recurrences with a high risk of metastasis and poor prognosis [4]. Despite advanced treatment strategies, the recommended treatment protocols are still under debate, and the survival of oral cancer patients has remained unchanged over recent years. In order to predict the prognosis of OSCC, many cell proliferation markers, tumor suppressor genes, angiogenesis, and adhesion molecules have been studied [5,6,7,8,9].

Heavily glycosylated transmembrane proteins, including mucin1 (MUC1), provide protection of epithelial linings of different organs from physical, chemical, and pathological threats [10]. In epithelial cancer cells, alteration of MUC1 O-glycosylation results in up to 10-fold overexpression of MUC1, loss of apical polarization, and its expression on the entire cell surface [11]. Such differential expression of MUC1 has been associated with tumor cell proliferation, migration, invasion, adhesion, and metastasis. MUC1 was shown to express homogeneously in the layers of irradiated tissue samples suggesting changes in the genome and microenvironmental properties of the individual cells that resulted in architectural and hierarchical disturbances in the epithelial cell strata, inter cellular interaction, and cell surface microstructure [12].

The Cluster of Differentiation (CD44) is a family of single-pass transmembrane glycoproteins known as a vital prognostic marker of several cancers [13,14]. CD44 is the key cell surface receptor for hyaluronan (HA). HA is an unbranched polysaccharide, which is produced by hyaluronan synthases 1–3 (HAS 1–3) at the plasma membrane of several cell types but is especially abundant in the connective tissue and skin epidermis [15]. HA binding to CD44 induces conformational changes that allow the binding of adaptor proteins or cytoskeleton elements to intracellular domains of CD44. This interaction induces intercellular signals, which influence cellular proliferation, migration, and invasion [16]. The binding ability of HA with CD44 is accelerated by the release of several inflammatory mediators [16,17]. Li et al. have shown in an animal study that irradiation causes transient accumulation of HA and impaired function of CD44 [18].

The expression of MUC1, CD44, and HA has been notably implicated in tumor cell migration and metastasis via several signaling pathways [11,17]. Previous publications have reported contradicting results—both upregulation and/or downregulation of MUC1, CD44, and HA have been connected to tumor progression. However, studies relating to their role in the pathophysiology of radiation-induced injury of the oral mucosa are sparse. Therefore, our aim was to determine the staining intensities of MUC1, CD44, and HA and their role in the long-term effects of RT in the irradiated oral mucosa.

## 2. Materials and Methods

### 2.1. Patient Characteristics

Oral mucosal tissue specimens were collected from a total of 59 fully edentulous patients who underwent dental implant surgeries in the anterior mandible. Group 1 (controls) consisted of 35 healthy patients without a history of oral cancer or RT. Group 2 (irradiated group) consisted of 24 patients with a positive history of RT to treat HNC. The interval between RT and biopsy was more than 11 months in irradiated patients. The exact local dose administered at the site of the biopsy was calculated by merging RT (intensity-modulated radiotherapy, IMRT) planning CT image with postoperative cone-beam CT-image. The demographics of the study group are presented in Table 1. The study was approved by the VU Medical Center Ethical committee (2011/220), Amsterdam, and the Research Ethics Committee of the Northern Savo Hospital District (754/2018; 21 April 2020). Tissue samples for the study were collected after all patients had signed a written informed consent to participate in the study, and the patient information was anonymous after sample harvesting. 

### 2.2. Sample Harvesting

The oral mucosal biopsies were taken from the anterior buccal vestibule of the mandibular ridge during dental implant surgery in both control and irradiated group. All subjects are edentulous and presented clinically normal-appearing oral mucosa. A tissue specimen of size 10 × 5 mm was obtained using a scalpel, fixed 10% phosphate-buffered formalin, and divided into two pieces, one for light microscopy (LM) and the other for electron microscopy. After fixation, the tissue was embedded in paraffin according to routine tissue processing. The tissue sections of 5 µm thickness were cut and stained first with hematoxylin-eosin (HE) for LM and thereafter immunohistochemical (IHC) staining for MUC1, CD44, and HA. 

#### 2.2.1. MUC1 Staining

MUC1 staining was performed using a Dako EnVision K 5007-kit (Dako, Glostrup, Denmark) as described in a previous study [12]. The method followed for IHC staining was as per the manufacturer‘s instructions (Abcam, Cambridge, UK). Briefly, for epitope retrieval, Tris-EDTA (pH 9) was used for 10 minutes at 98 °C (Milestone). Dako Peroxidase-Blocking solution S2023 was used for 15 min to remove endogenous peroxidase to block non-specific staining. Samples were incubated with primary antibody for MUC 1 (concentration 2 µg/mL, 1:500 dilution in Dako REAL Antibody diluent S2022) for 1 h at room temperature. MUC1 monoclonal antibody, human milk fat globule 1 (HMFG 1), from Abcam (Cambridge, UK), was used. Labeled polymer horseradish peroxidase (HRP) was used for 30 min with a conjugated secondary antibody at room temperature. DAB as chromogen was used for 5 min. The sections were subsequently washed with distilled water, counterstained with Mayer’s hematoxylin for 1 min, and blued in ammonia water solution (0.75%) for 1 min. A PBS-Tween wash was used in between the staining protocols. For negative control, we used NIS Dako mono, and for positive control, we used a sample of minor salivary glands with strong staining of MUC1 in the mucinous glands. 

#### 2.2.2. HA Staining

The tissue sections (5 µm) were first kept at 58 °C for 30 min, after which deparaffinization was accomplished by incubating in 1% H_2_O_2_ for 5 min to block endogenous peroxidase. Sections were washed with water and phosphate buffer (PB). Non-specific binding of the antigen was blocked by incubating in 1% bovine serum albumin (BSA) in PB at 37 °C for 30 min. The sections were incubated overnight at 4 °C in 1:30 dilution of 50 µL/mL biotinylated hyaluronan binding complex (bHABR), prepared as described by Tammi et al. [19]. The sections were washed with PB, and the bound HA binding complex was visualized with the avidin-biotin-peroxidase (ABC) method (1:200, Vectastain ABC Kit, Vector Laboratories, Burlingame, CA, USA) for 1 h at room temperature. The sections were subsequently washed with PB. Diaminobenzidine (DAB; Sigma, St. Louis, MO, USA) was used as chromogen, containing 0.5 mg/mL PB and 0.03% H_2_O_2_, incubating for 5 min. The sections were washed with distilled water and counterstained with Mayer’s hematoxylin for 1 min. The sections were washed, dehydrated, and mounted in DPX (BDH Laboratory Supplies, Poole, UK). To control the specificity of the staining, the specimens were predigested with *Streptomyces* hyaluronidase (100 TRU/mL in acetate buffer, pH 5.0 for 3 h at 37 °C; Seikagaku, Kogyo, Tokyo, Japan) in the presence of protease inhibitor.

#### 2.2.3. CD44 Staining

In order to stain CD44, sections were deparaffinized, rehydrated, and washed with PB. Endogenous peroxidase was blocked by 1% H_2_O_2_ for 5 min followed by washing with PB. The sections were incubated overnight at 4 °C in 1:200 dilution of Hermes-3 antibody against CD44 in BSA solution. Sections were then washed twice with PB and treated with biotinylated secondary anti-mouse antibody (Vector Laboratories) at a dilution of 1:150 PB for 1 h. The sections were washed with PB and incubated for 40 min in a preformed ABC complex, washed, and color developed with DAB and H_2_O_2,_ as described above. The sections were dehydrated, cleared, and mounted in DPX. Negative controls were prepared similarly but omitted the Hermes-3 antibody.

### 2.3. Staining Analysis and Statistics

Two researchers performed the examination and scoring of MUC1, CD44, and HA staining under LM. An ordered recording was carried out for staining distribution in the basal, intermediate, and superficial layers of the oral mucosal epithelium, as shown in Figure 1. The staining intensity for MUC1, CD44, and HA was estimated as described previously: 0 = no staining, 1 = some staining, 2 = considerable staining [12].

The obtained results were statistically analyzed using IBM SPSS statistic version 27 for Windows (SPSS, Chicago, IL, USA). Staining intensity among groups was analyzed using cross-tabulation and the Fischer-Freeman-Halton test, with a *p*-value of <0.05 being considered statistically significant. The Chi-square test was performed to analyze the correlation between radiation dose and biopsy time interval, and the statistical significance was set as *p* < 0.05.

### 2.4. Transmission Electron Microscopy (TEM)

Transmission electron microscopy was used to investigate cell-cell interaction with higher magnification. The samples were post-fixed in 2% osmium tetroxide with a cacodylate buffer solution at 4 °C for two hours before rinsing with cacodylate buffer solution, dehydrating in graded ethanol, and embedding in epoxy resin as described previously [12]. Semi-thin sections were made with an ultramicrotome and stained with 1% toluidine blue and examined with a light microscope to find a correct area for TEM investigations. After that, semi-thin sections were stained with uranyl acetate and investigated with a JEOL JEM-2100F transmission electron microscope equipped with a digital camera (Olympus-SIS, Munster, Germany).

## 3. Results

Under light microscopy, all the oral mucosal samples showed non-keratinized stratified squamous epithelium supported by underlying connective tissue stroma. Some of the samples were presented with thin and broad rete ridges, and few were without rete ridges. The underlying connective tissue stroma demonstrated inflammatory cells in some of the Group 2 samples. 

### 3.1. The Intensity of MUC1, CD44 and HA in Control and Irradiated Mucosa

In control samples, MUC1 staining was limited to the apical part of the cells of the superficial layer of oral epithelium, whereas the basal cell layer and subepithelial connective tissue were negative for MUC1. The HA staining was observed in all the epithelial layers of controls except the most superficial layer, which was negative in most of the samples (71%). HA staining intensity was higher in the basal layer, followed by the intermediate layer. The CD44 staining pattern was identical to that of HA staining in controls.

The specimens of irradiated oral mucosa showed MUC1 staining in all the epithelial cell layers, with considerable staining in the basal (38%) and intermediate layers (71%). A comparison of the MUC1 staining intensity among controls and irradiated mucosa revealed an increased staining intensity in the basal and intermediate layers (*p* < 0.0001 and *p* < 0.001, respectively). The MUC1 staining in the superficial layer of the irradiated mucosa did not show any difference compared to the controls. Irradiated sections showed that HA positivity extended up to the most superficial layers. Both HA and CD44 in the irradiated sample showed membranous staining, with a few cells showing intracellular staining in the intermediate and superficial layers of epithelium, along with considerable staining in the subepithelial connective tissue. In irradiated specimens, the HA staining was significantly increased in the stroma (*p* < 0.01), the intermediate (*p* < 0.0001), and the superficial layers (*p* < 0.001) when compared to controls. Furthermore, CD44 expression was consistently increased in the intermediate layer of the irradiated specimens (*p* < 0.01) compared to the controls (Table 2).

### 3.2. Immunostaining of HA and CD44 in Irradiated Mucosa

HA and its CD44 receptor showed identical, either homogeneous or irregular, staining patterns among groups. The control group showed membranous expression of HA and CD44 in the basal cells and intermediate layer of the epithelium, along with considerable staining of the stroma. In control samples, the basal layer of the epithelium showed some considerable staining of HA and CD44, but the superficial layers were negative. A distinct difference from the normal staining pattern was evident in irradiated samples. The HA and CD44 positivity extended to involve membranous and intracellular staining of the most superficial layers, showing a significant *p*-value in the superficial layers (*p* < 0.01) of irradiated sections compared to controls (Figure 2). Further, the correlation of HA and CD44 with the local radiation dose and the time interval between radiation dose and biopsy did not show any difference. 

### 3.3. MUC1 and CD44 Atypical Immunostaining in Irradiated Mucosa

The precise location of the MUC1 in the apical portion of cells of the superficial layer and CD44 in the membranes of cells of basal and intermediate layers is well evidenced in our control group. The staining pattern of MUC1 was increased (*p* < 0.001) in the superficial layer and decreased in the basal layer (*p* < 0.002) in control samples compared to CD44 staining in the superficial and basal cell layers. MUC1 and CD44 presented changes in staining patterns in irradiated tissue samples. MUC1 expression is observed in the apical, basal, and lateral portions of the epithelium cells. CD44 was expressed clearly in the plasma membrane and intracellular area of the intermediate cell layer, with some samples expressing CD44 in the superficial cell layers of the oral epithelium in the irradiated samples. Only one irradiated sample (4%) showed considerable MUC1 staining in the stroma, whereas CD44 showed considerable staining in 20 samples (83%) and 4 samples (17%) showed negative staining. MUC1 and CD44 showed an increased difference in the stromal staining (*p* < 0.008) and superficial layers of the irradiated epithelium (*p* < 0.0001) (Table 2). 

### 3.4. Cell Junction Alteration in the Suprabasal Cell Layer

In controls, MUC1 expression was low, and it localized to the apical cell membrane of suprabasal and intermediate cells. In irradiated epithelium, the MUC1 was localized to the cell surface in the superficial, intermediate, and basal cell layers (Figure 3A,D). HA was detected intercellularly and in the plasma membrane of controls, whereas the irradiated epithelium showed localization of HA at the plasma membrane with the presence of small vacuolated areas disrupting the continuity of the plasma membrane (Figure 3B,E). In TEM examination, regular cell-to-cell interaction and appropriate cell junctions were observed between adjacent cells in the suprabasal cell layer of controls. An asymmetrical disrupted or disordered cell junctional component was observed in the suprabasal cell layer of the irradiated sample (Figure 3C,F). 

## 4. Discussion

The present study displays for the first time the immunohistochemical localization and staining pattern of MUC1, HA, and CD44 in the irradiated oral mucosa. The results indicate the altered and increased staining intensity of MUC1, CD44, and HA in irradiated mucosal tissue, compared to the normal oral mucosal tissue. 

The distribution of HA and CD44 in oral mucosa appears to be homogeneous and strong in the basal and intermediate layers and absent in the most superficial layers. A similar distribution is also seen in other stratified epithelia such as gingiva, epidermis, laryngeal epithelium, and esophagus [19,20,21]. While MUC1 is expressed in the apical border of oral mucosal epithelial cells, as reported previously [10,22]. A decrease in HA and CD44 staining intensity or irregular staining with focal reduction in HA and CD44 have previously been reported in squamous cell carcinoma [15,17,18,20]. However, the HA and CD44 staining patterns were demonstrated to increase in our irradiated study group in the intermediate and superficial layers. This irregular staining was frequently observed pericellular, with few cells showing intracellular staining. These differences in the intensity and localization of HA and CD44 expression in the irradiated tissues reflect a similar finding to that observed in OSCC, which includes the altered balance between synthesis and catabolism of HA, increased uptake of CD44-bound HA, delivery to lysosomes, change in CD44 phosphorylation, change in cytoskeletal binding proteins or extracellular proteolytic activity [18]. The cell-cell and cell-matrix interactions are mediated by HA and CD44, and if downregulated, the ability of the cells to escape the epithelium and invade the surrounding tissues is promoted [23]. Our results showed that plasma membrane and intracellular staining of HA and CD44 extended up to involve the superficial layers of epithelium, possibly due to co-localization of HA and CD44 in irradiated tissue or upregulation of HA and CD44 production due to some persistent inflammatory stimulus following RT. Moreover, HA and CD44 expression did not correlate with the local dose of radiation and the time interval between RT and biopsy, such as the MUC1 expression, as shown previously [12].

MUC1 is expressed in the apical portion of the superficial layer of normal epithelial cells [12]. In dysplastic epithelium and cancers, MUC1 is detected over the entire cell surface [24]. We observed that MUC1 is expressed in all the layers of irradiated tissue samples, as shown in a previous paper [12]. The MUC1 expression in basal, intermediate, and superficial layers of the oral epithelium could hypothetically explain the aberrant MUC1 recycling process, as suggested by Litvinov and Hilkens [25]. The overexpression and distribution of MUC1 on cell surfaces, as well as in the cytoplasm of the transformed cell, due to changes in cell polarity, reduces cell-cell and cell-matrix interactions, thereby promoting epithelial cell motility and migration [26]. However, the MUC1 expression did not correlate with the radiation dose and biopsy interval, but the long-term changes are evident even 10 years after RT.

The extracellular, transmembrane, and cytoplasmic domains of the MUC1 play varied roles in cancer progression [24]. The upregulation and repositioning of MUC1 over the entire cell surface allow it to form complexes with many cell surface receptors positioned at basal, lateral, or apical borders and direct the activation of multiple signaling pathways [14]. In this study, we observed the staining pattern of MUC1 and CD44 and compared it among the study groups. To our surprise, MUC1 and CD44 were distributed in the entire cell layers of the irradiated group, more significantly in the superficial and basal cell layers. It is proposed that during cancer progression, there is partial to complete alteration in the differentiation pattern of keratinocytes which results in loss of polarity [27]. Hence, we believe that also in irradiated tissue, MUC1 and CD44 expression would be induced by loss of cell polarity and alteration in the differentiation of keratinocytes. 

Long-standing epithelial changes occur at the molecular and cellular level following RT. The radiation targets the rapidly proliferating cells in the basal cells, thus impairing the ability of the tissue to renew itself [28]. The mitotic activity in the basal cell decreases, resulting in a reduced and altered turnover of the keratinocytes of the oral mucosa. The radiation-induced inflammation increases the hyaluronan synthesis and release of inflammatory mediators such as transforming growth factor-beta (TGFβ) and Interleukin 1 (IL1), which in turn increases the binding activity of CD44 and hyaluronan [17]. The increased expression of hyaluronan suggests the affinity of CD44-hyaluronan binding following radiation. The rupture of cell junctions and altered extracellular links in the epithelial cells of irradiated oral mucosa was demonstrated by transmission electron microscopy. Similarly, irradiation also altered MUC1 expression, causing loss of apical-basal polarity of oral epithelial cells. The protective barrier provided by MUC1 was affected, leading to a reduced ability of the tissue to heal and resist infection. Hence, the long-term persistent infection may limit the proliferative ability of the epithelium and cause ulceration of the superficial layer. The clinical manifestations of radiation, such as erythema, edema, inflammation, and ulceration, imply that the proliferating oral epithelium is more sensitive to radiation than the underlying connective tissue of the oral mucosa [29]. The regenerative epithelial response, together with intrinsic radiosensitivity, could be the dominant parameter of the radiation tolerance of the oral mucosa. Despite changes occurring at the subcellular level in the oral epithelium, there is still some functional adaptation taking place by keratinocytes as an indirect effect of radiation, resulting in the absence of any clinical signs.

A variety of intra- and intercellular communication pathways might be modulated in the irradiated oral mucosa, such as nuclear factor (NF-κB), reactive oxygen species (ROS), tumor protein 53 (p53), nuclear factor erythroid-related factor 2 (NRF2), and JNK. Similarly, interactions with multiple effectors such as PI3K, NF-kB, p65, and β-catenin promote anchorage-independent growth and transformation [24,26,27,28,30]. Based on the observed immunostaining distribution in the irradiated tissue, we assume that the cytoplasmic domain of MUC1 might interact with the CD44 intracellular domain. Such interaction might impact cellular and molecular changes via the extracellular signal-regulated kinase (ERK) pathway, the phosphatidylinositol 3-kinase (PI3K) pathway, or the Janus/Signal transducer and activator of transcription (JAK/STAT) pathway [11,31]. These interactions and accompanied extra-epithelial changes such as epithelial atrophy, ulceration, and regeneration currently remain complex and largely unclear. 

Overall, we showed the long-term changes after RT in oral mucosa using immunostaining of MUC1, HA, and CD44, although the healing process of oral mucosa occurs about 3 weeks after RT [28]. The destroyed cell-cell interaction of the oral epithelium might be a reason for easy microbial invasion causing oral lesions years after RT. MUC1 and CD44 expression might be a valuable tool in assessing patients’ risk of progression and/or of developing radiation-induced lesions, but other factors, such as the site of the biopsy and smoking, must be taken into consideration in future studies.

## Figures and Tables

**Figure 1 biomedicines-10-02816-f001:**
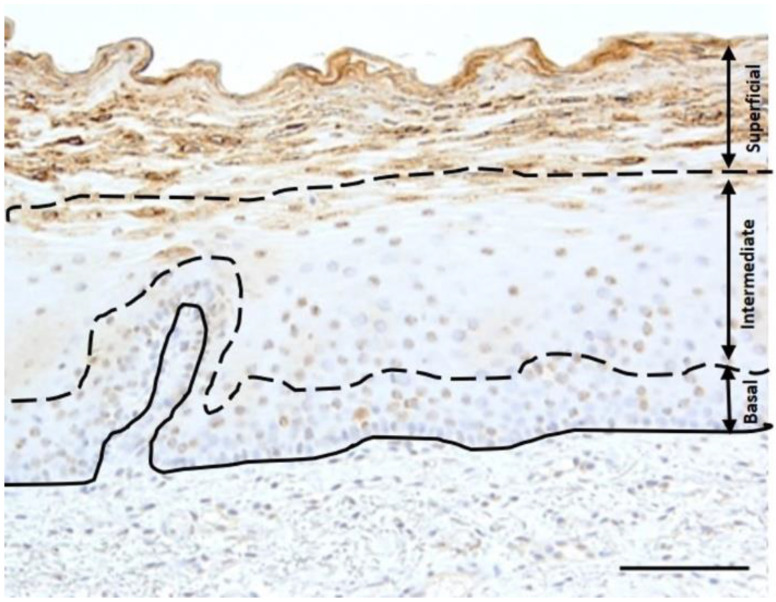
The scoring of the staining intensity and the immune localization of MUC1, CD44, and HA in the epithelium was evaluated in the basal cell layer, intermediate cell layer, and superficial cell layer under a light microscope (Bar = 100 µm).

**Figure 2 biomedicines-10-02816-f002:**
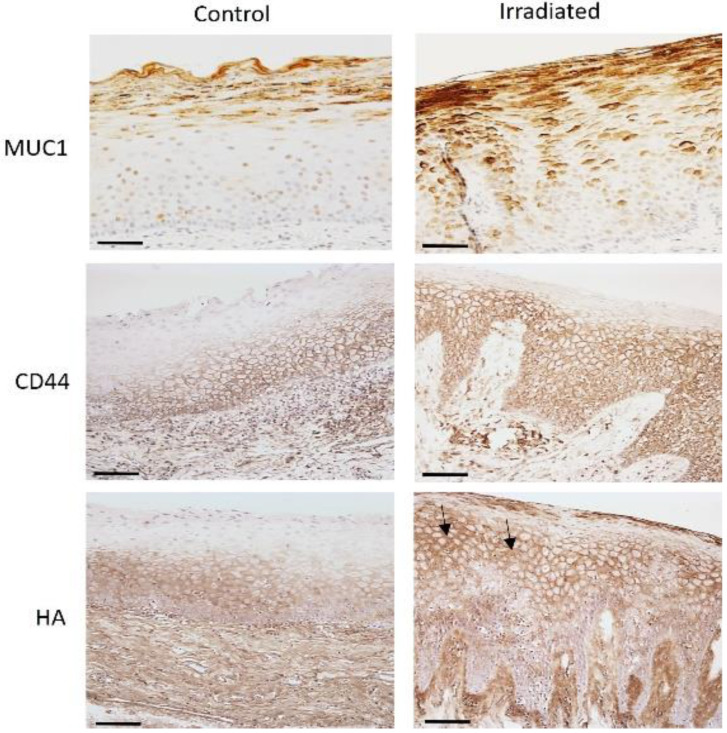
MUC1 is expressed in the superficial layers of the control oral epithelium. Increased expression of MUC1 showing membrane and cytoplasm staining in all the layers of epithelium is observed in the irradiated epithelium. Immunolocalization of HA and CD44 is limited to pericellular and faint intracellular staining in basal and intermediate layers of controls. Immunohistochemistry images of HA and CD44 show a gradual increase in staining intensity in irradiated epithelium together with peri- and intracellular staining. The intermediate cells show cytoplasmic HA staining (arrows) in the irradiated epithelium (Bar = 100 µm).

**Figure 3 biomedicines-10-02816-f003:**
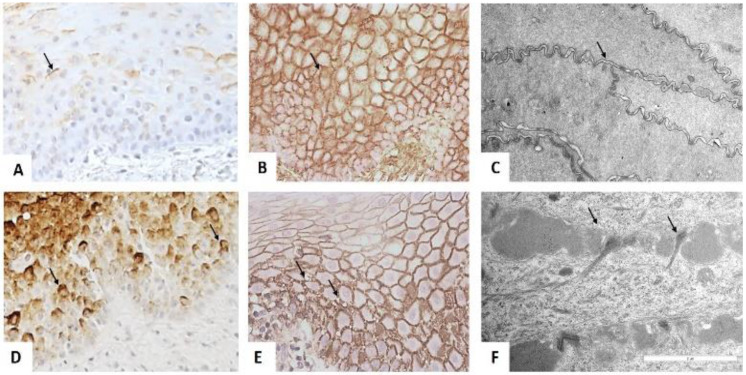
In controls, MUC1 expression is faint in the apical surface of basal and supra-basal cell layers (arrow) (Bar = 200 µm) (**A**). In the irradiated sample, MUC1 expressed involving all the apical, lateral, and basal surfaces of the basal and supra-basal cells (arrows) (Bar = 200 µm) (**D**). HA is expressed on the plasma membrane of control (Bar = 200 µm) (**B**). The HA staining in the irradiated epithelium at the plasma membrane is clear and shows the presence of numerous vacuoles (arrows) (Bar = 200 µm) (**E**). Transmission electron microscopic images showing cell junctions are regularly arranged in controls (**C**), whereas in irradiated, the intercellular junctions are disrupted and broken (arrows) (**F**) (Bar = 2 µm).

**Table 1 biomedicines-10-02816-t001:** Demographics of study patients, tumor site, and treatment characteristics.

Patients (*N* = 59)	Male/Female	Age Range (Mean Age)	Tumor Site (*N*)	Total RT Dose (Gy) (Mean Dose)	Local RT Dose * (Gy) (Mean Dose)	Interval RT Biopsy (Months)
**Group 1** Controls (*N* = 35)	16/19	33–79 y (56 y)	-	-	-	-
**Group 2** (*N* = 24)	20/4	54–84 y (69 y)	OC (*N* = 14) SG (*N* = 5) NP or L (*N* = 5)	54–70 (62 Gy) 54–70 Gy 60–70 Gy 70 Gy	15–66 (41 Gy) 34–66 Gy 32–60 Gy 15–34 Gy	11–199 11–199 10–23 72–171

Abbreviations: RT = radiation therapy; * Maximum radiation dose at biopsy site; OC = oral cavity; SG = salivary glands; NP = Nasopharynx; L = larynx.

**Table 2 biomedicines-10-02816-t002:** The percentage for staining intensity of MUC1, CD44, and HA as considerable staining, some staining, and no staining are assessed in basal, intermediate, and superficial cell layers of oral epithelium and in stroma among control and irradiated groups. The statistical significance of staining intensity between controls and irradiated (IR) was tested using the Fischer-Freeman-Halton test.

	MUC1		CD44		HA	
	Control (*n* = 35)	IR (*n* = 24)	Control (*n* = 35)	IR (*n* = 24)	Control (*n* = 35)	IR (*n* = 24)
**Superficial cells**						
-Considerable staining	83%	75%	3%	33%	3%	29%
-Some staining	17%	13%	26%	38%	26%	38%
-No staining	0%	13%	71%	29%	71%	33%
				*p* < 0.01		*p* < 0.01
**Intermediate layer**						
-Considerable staining	31%	71%	17%	86%	17%	58%
-Some staining	57%	13%	50%	14%	46%	42%
-No staining	11%	17%	33%	0%	37%	0%
		*p* < 0.001		*p* < 0.001		*p* < 0.0001
**Basal layer**						
-Considerable staining	0%	38%	66%	71%	34%	38%
-Some staining	0%	46%	33%	29%	54%	54%
-No staining	100%	17%	0%	0%	11%	8%
		*p* < 0001				
**Stroma**						
-Considerable staining	0%	4%	60%	40%	60%	29%
-Some staining	0%	0%	40%	43%	40%	54%
-No staining	100%	96%	0%	17%	0%	17%
						*p* < 0.01

## Data Availability

All data analyzed and generated in this study are included within the article, or any specific dataset can be obtained from the corresponding author on reasonable request.

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
