# Peer review of "Irradiation Alters the Expression of MUC1, CD44 and Hyaluronan in Oral Mucosal Epithelium"

_biomedicines, 2022, doi:10.3390/biomedicines10112816_

Round 1

Reviewer 1 Report

This manuscript that submitted by Kashyap et al. aimed to evaluated the expression level of CD44 and hyaluronan (HA) after patients receiving radiotherapy. Their results showed that the staining pattern of MUC1 and CD44 significantly changed in irradiated samples. There are some suggestions for the authors.

1. All results that showed in this manuscript were too small to observe. The authors must enlarge all figures which present in this manuscript.

2. The IHC staining data should provide the table for summarizing each protein signaling intensity and distribution with statistical analysis.

3. The CD44 had several splicing proteins, the authors should provide the detail of MUC1, CD44 and HA antibodies information. In addition, the corresponding HA receptor for CD44 must be included in IHC analysis.

4. It is expected that extracellular matrix would occur remodeling after receiving the radiotherapy. I am curiously why the authors decide to examine the expression level of CD44 and HA. Although the authors provided refs. aiming to emphasize their influence in oral cancer. But, these refs didn't figure out any clinical significance in patients outcome? How about the expression level of MUC1, CD44 and HA on patients survival rate?  In addition, the collagen is highly abundance in oral tissues, so, how about the expression level of HA on oral cancer patients survival rate compared with other types of ECM? Overall, the present study only provided the proteins expression level in cancer tissues, but it lacked the clinical significant analysis.

5. The radiotherapy is a standard therapy for head and neck cancer patients, I am not sure if this study aims to claim the disadvantage on radiotherapy? However, most oral cancer patients that received the radiotherapy combining with chemotherapy eventually had higher disease-free survival rate. The impact of the present study should be further considerations.

Author Response

We acknowledge the reviewer’s comments. Please, see the attached file.

This manuscript that submitted by Kashyap et al. aimed to evaluated the expression level of CD44 and hyaluronan (HA) after patients receiving radiotherapy. Their results showed that the staining pattern of MUC1 and CD44 significantly changed in irradiated samples. There are some suggestions for the authors.

  1. All results that showed in this manuscript were too small to observe. The authors must enlarge all figures which present in this manuscript.

Author response: As per the reviewer’s comment, all the figures are enlarged and have been checked for the clarity and the dpi (300). 

  1. The IHC staining data should provide the table for summarizing each protein signaling intensity and distribution with statistical analysis.

Author response: The IHC data was presented in Figure 2, but as per the reviewers comment the figure 2 has been replaced with table 2 (Under section 3.1 ), summarizing the MUC1, CD44 and HA staining intensity and the statistical analysis. 

  1. The CD44 had several splicing proteins, the authors should provide the detail of MUC1, CD44 and HA antibodies information. In addition, the corresponding HA receptor for CD44 must be included in IHC analysis.

Author response: We appreciate the reviewers comment. We have provided the information of antibodies used in our study in following lines: line no 124 for MUC1 (HMFG1), line no 151 for HA (bHABR), and line no 165 for CD44 (Hermes-3).  

  1. It is expected that extracellular matrix would occur remodeling after receiving the radiotherapy. I am curiously why the authors decide to examine the expression level of CD44 and HA. Although the authors provided refs. aiming to emphasize their influence in oral cancer. But, these refs didn't figure out any clinical significance in patients outcome? How about the expression level of MUC1, CD44 and HA on patients survival rate?  In addition, the collagen is highly abundance in oral tissues, so, how about the expression level of HA on oral cancer patients survival rate compared with other types of ECM? Overall, the present study only provided the proteins expression level in cancer tissues, but it lacked the clinical significant analysis.

Author response: In section 2.1 Patients characteristics, we have presented the inclusion criteria and the material does not include any cancer tissue. We don’t correlate survival rate with expression level of biomarkers. All subjects have clinically normal-appearing oral mucosa at the time of dental implant surgery.

  1. The radiotherapy is a standard therapy for head and neck cancer patients, I am not sure if this study aims to claim the disadvantage on radiotherapy? However, most oral cancer patients that received the radiotherapy combining with chemotherapy eventually had higher disease-free survival rate. The impact of the present study should be further considerations.

Author Response: We acknowledge the reviewer’s comment. Our study does not aim on the disadvantage of radiotherapy as all the patients included in our study had undergone for oral mucosal biopsy during dental implant surgery. This suggest that all our patients had a good recovery after IMRT radiotherapy of head and neck cancers at least 11 months.

Reviewer 2 Report

Dear authors, this is a rather novel study that provides a new insight of tissue expression after head and neck radiotherapy.

Nevertheless, there are some issues in methodology that I would like to address to. First of all, the type of study must be identified. Is this a retrospective or prospective study? If the first is the case. The patient selection must be identified. Either way, Patient inclusion and exclusion criteria must be set. In addition, what is rather important in patients who have received had a neck radiotherapy is the time that has lapsed prior to the excision of the bioptic material. It It must be also also stated that the retromolar area has been included in the radiotherapy field so as to justify the biopsy of that area as irradiated oral mucosa.

Author Response

We acknowledge the reviewer’s comments.

Dear authors, this is a rather novel study that provides a new insight of tissue expression after head and neck radiotherapy.

Nevertheless, there are some issues in methodology that I would like to address to. First of all, the type of study must be identified. Is this a retrospective or prospective study? If the first is the case. The patient selection must be identified. Either way, Patient inclusion and exclusion criteria must be set. In addition, what is rather important in patients who have received had a neck radiotherapy is the time that has lapsed prior to the excision of the bioptic material. It It must be also also stated that the retromolar area has been included in the radiotherapy field so as to justify the biopsy of that area as irradiated oral mucosa.

Author response: We are thankful to the reviewer’s comment. This is a retrospective study, and the selection of the patient has been mentioned under section 2.1 patients’ characteristics, line number 90-94. The details of the type of cancer, local dose of radiation and biopsy interval are presented in Table 1. Also, the excision of the biopsy sample has been mentioned under section 2.2 sample harvesting, line number 109-111.

Reviewer 3 Report

I would like to congratulate the authors for going forward with their recent research. The title of the article is “Irradiation alters the expression of MUC1, CD44 and hyaluronan in oral mucosal epithelium”. It samples 35 patients in the control group and 24 in the irradiated group to compare the MUC1, CD44 and HA. The results showed major differences in the expression of the receptors. I found this research article to be quite well rounded, with a good research hypothesis and good design. I have still some minor remarks which may help the authors improve on some aspects: 1. In the abstract it is not very clear the materials and methods: I would recommend including the method of sampling (oral mucosa form patients and so on ) 2. In the text in the materials and methods: it is not clear whether the samples were taken during dental implant surgery insertion or afterwards. 3. The approval of research usually has a number and date to it 4. The sample harvesting: I do not exactly understand what type of tissue was harvested? Attached/ fixed gingiva or mucosa from the vestibule? 5. All the figures need explanation 6. Also figure 2 is quite small and hard to read (the attached figures seem to be large enough) 7. In figures where there are arrows the authors should help the reader understand what those arrows try to represent

Author Response

We acknowledge the reviewer’s comments.

I would like to congratulate the authors for going forward with their recent research. The title of the article is “Irradiation alters the expression of MUC1, CD44 and hyaluronan in oral mucosal epithelium”. It samples 35 patients in the control group and 24 in the irradiated group to compare the MUC1, CD44 and HA. The results showed major differences in the expression of the receptors. I found this research article to be quite well rounded, with a good research hypothesis and good design. I have still some minor remarks which may help the authors improve on some aspects:

  1. In the abstract it is not very clear the materials and methods: I would recommend including the method of sampling (oral mucosa form patients and so on ).

Author response: We are thankful to the reviewer’s comment. As per the reviewer’s comment- we have modified the materials method section in Abstract.

  1. In the text in the materials and methods: it is not clear whether the samples were taken during dental implant surgery insertion or afterwards.

Author response: With respect to the reviewer’s comment, author would like to point that in section 2.2 sample harvesting, line number 109-111, it is mentioned that the biopsy were retrieved during dental implant surgery.

  1. The approval of research usually has a number and date to it

Author response: We are thankful to the reviewer’s comment. As per the reviewer’s comment- we have included the approval of research with number assigned, under section 2.1 Patients characteristics.

  1. The sample harvesting: I do not exactly understand what type of tissue was harvested? Attached/ fixed gingiva or mucosa from the vestibule?

Author response: We are thankful to the reviewer’s comment. Authors have mentioned that the biopsies were taken from the anterior buccal vestibule during dental implant surgery.  Menthioned in section 2.2. Sample harvesting.

  1. All the figures need explanation

Author response: As per the reviewer’s comment, legends to all the figures have been added.

  1. Also figure 2 is quite small and hard to read (the attached figures seem to be large enough)

Author response: As per the reviewer’s comment, Figure 2 is replaced with the table showing the staining intensities and the statistical analysis of the biomarkers used.

  1. In figures where there are arrows the authors should help the reader understand what those arrows try to represent

Author response: As per the reviewer’s comment, the figure is enlarged, and the legend is added for better appreciation.

Round 2

Reviewer 1 Report

The authors have responded all questions and make revision to let this manuscript become better. It seems to reach the criterion for publishing.

Reviewer 2 Report

Dear authors, thank you for your response. All questions have been adequately answered.